# Grain Growth Kinetics of 0.65Ca_0.61_La_0.26_TiO_3_-0.35Sm(Mg_0.5_Ti_0.5_)O_3_ Dielectric Ceramic

**DOI:** 10.3390/ma13173905

**Published:** 2020-09-03

**Authors:** Jin Liu, Bingliang Liang, Jianjun Zhang, Wen He, Sheng Ouyang, Weihua Chen, Changhong Liu, Yunlong Ai

**Affiliations:** 1Key Laboratory for Microstructural Control of Metallic Materials of Jiangxi Province (Nanchang Hangkong University), Nanchang 330063, China; 1801085204026@stu.nchu.edu.cn (J.L.); zhangjianjun71@nchu.edu.cn (J.Z.); hewen@nchu.edu.cn (W.H.); 70685@nchu.edu.cn (S.O.); chenweihua@nchu.edu.cn (W.C.); 27014@nchu.edu.cn (C.L.); ayunlong@126.com (Y.A.); 2School of Materials Science and Engineering, Nanchang Hangkong University, Nanchang 330063, China

**Keywords:** CLT-SMT ceramic, growth models, nonlinear regression, dielectric materials

## Abstract

The 0.65Ca_0.61_La_0.26_TiO_3_-0.35Sm(Mg_0.5_Ti_0.5_)O_3_[0.65CLT-0.35SMT] ceramic was prepared by the solid-state reaction method. The effects of sintering process on its microstructure and grain growth behavior were investigated. The Hillert model and a simplified Sellars model were established by linear regression, and the Sellars-Anelli model with a time index was established by using a nonlinear regression method. The results show that the grain size gradually increases with the increase of sintering temperature and holding time. Meanwhile, the sintering temperature has a more significant effect on the grain growth. The grain sizes of 0.65CLT-0.35SMT ceramic were predicted by the three models and compared with the experimentally measured grain size. The results indicate that for the 0.65CLT-0.35SMT ceramic, the Hillert model has the lowest prediction accuracy and the Sellars-Anelli model, the highest prediction accuracy. In this work, the Sellars-Anelli model can effectively predict the grain growth process of 0.65CLT-0.35SMT ceramic.

## 1. Introduction

In recent years, with the rapid development of microwave mobile communication technology, especially the advent of the 5G commercial era, microwave dielectric ceramics has become a research hotspot worldwide. Excellent performance microwave dielectric ceramics have the following properties: Moderate dielectric constant *ε_r_*, high quality factor *Qf*, and near-zero resonance frequency temperature coefficient τ*_f_*. According to the characteristics of microwave dielectric properties, microwave dielectric ceramics can be divided into low *ε_r_* (*ε_r_* < 15) and high *Qf* ceramics: Al_2_O_3_ [1], MgAl_2_O_4_ [2], etc.; medium *ε_r_* (*ε_r_*~30) and *Qf* value ceramics: BaTi_4_O_9_ [3], ZnNb_2_O_6_ [4], etc.; high *ε_r_* (*ε_r_* > 60) and low *Qf* materials: CaCu_3_Ti_4_O_12_ [5], Li_2_O-Nb_2_O_5_-TiO_2_ [6], etc. Various types of microwave dielectric ceramics with different performances are widely used in the production of dielectric resonators, dielectric filters, dielectric oscillators, phase shifters, microwave capacitors, microwave substrates, and other microwave components. Microwave dielectric ceramics play an increasingly important role in the miniaturization, integration, and cost reduction of modern communication tools [7,8,9,10,11,12].

Grain size has an important influence on the properties of various materials [13,14]. Through the study of the grain growth kinetics, a grain growth kinetic model can be constructed to analyze the growth rate and growth mechanism of the grain. This provides favorable conditions for controlling the grain size and crystal structure to achieve the desired performance. Therefore, more attention has been paid to the research on the kinetics of grain growth. Qing et al. [15] studied the grain growth behavior of low-carbon Nb-V-Ti microalloyed steel X70, and established three grain growth models of the steel using the Beck, Hillert, and Sellars models. With a high prediction accuracy, the Sellars model can effectively predict the grain growth behavior of X70 steel. The grain growth kinetics model can be applied not only to metallic materials but also to ceramic materials. Du et al. [16] calculated the kinetic index and growth activation energy of *x*Nb_2_O_5_-7.5La_2_O_3_-Al_2_O_3_ (*x* = 7.5, 10, 12.5, 15) ceramics using the simplified Sellars model, and analyzed the columnar grains growth regularity and interface reaction process.

The effects of sintering temperature and holding time on the grain size growth law of 0.65CLT-0.35SMT ceramics were studied. Hiller model, simplified Sellars model, and Sellars-Anelli model were established, respectively. The three models were compared to predict the accuracy of the ceramic grain growth. A kinetic model that conforms to the grain growth of this ceramic was obtained, thus providing theoretical guidance for the sintering process of this ceramic.

## 2. Experimental Process

The 0.65CLT-0.35SMT ceramics were prepared by the solid phase reaction method. The ingredients were proportioned according to the stoichiometric ratio. High-purity CaCO_3_ (99.8%), La_2_O_3_ (99.9%), Sm_2_O_3_ (99.9%), MgO (99.99%), and TiO_2_ (99.5%) powders were mixed by a ball mill for 8 h and then dried, ground, sieved, and calcined (CLT at 1200 °C for 3 h, SMT at 1400 °C for 3 h, respectively). Then, the calcined CLT and SMT powders were ball milled, dried, and sieved (200-mesh). After adding about 10 wt% of polyvinyl alcohol solution (PVA, 10%) as a binder, the mixed 0.65CLT-0.35SMT powders were pressed into columns with a diameter of 13 mm and a thickness of 2~6 mm and then these specimens were heated at 600 °C for 1 h to remove the PVA. Finally, these specimens were sintered in air in a box-type electric furnace (Luoyang Hengyu experimental electric furnace Factory, Luoyang, China) with a heating rate of 10 °C/min (1525~1600 °C, 30~240 min).

After being polished and ultrasonically cleaned, the sintered 0.65CLT-0.35SMT ceramic specimens were thermally corroded at 25 °C below the sintering temperature for 20 min. The morphologies of the mixed powders and sintered specimens were observed with a scanning electron microscope (SEM, FEI, Hillsboro, OR, USA). The initial particle size of the calcined 0.65CLT-0.35SMT powders and grain size of 0.65CLT-0.35SMT ceramic specimens were measured by using the Nano Measurer 1.2 software (Fudan University, Shanghai, China), with 100 particles or grains for each specimen.

## 3. Results and Discussion

### 3.1. Microstructure Analysis

The SEM image of mixed 0.65CLT-0.35SMT powders is shown in Figure 1. It shows that most of the 0.65CLT-0.35SMT powders are spherical and a few of them are occasionally agglomerated. Meanwhile, the average size of the mixed 0.65CLT-0.35SMT powders is measured as 1.31 μm.

Table 1 shows the average grain size (*d*) and density (*η*) of 0.65CLT-0.35SMT ceramics obtained at different temperatures (*T*) for different holding times (*t*). As listed in Table 1, *d* gradually increases with increasing *T* or *t*. The maximum *d* of 0.65CLT-0.35SMT ceramic is 44.49 μm, which is about 34 times of the initial average particle size (*d*_0_) of mixed 0.65CLT-0.35SMT powders (1.31 μm).

### 3.2. Effects of Sintering Temperature and Holding Time on Grain Growth

Figure 2 demonstrates the effects of sintering temperature and holding time on the average grain size of 0.65CLT-0.35SMT ceramics. In Figure 2a, the slopes of the four lines are similar in the temperature range of 1525~1550 °C, indicating that the growth rate of grains is relatively similar at different holding times. In addition, the largest slopes in the temperature range of 1550~1575 °C indicate that the grain growth is accelerated in this temperature range. Figure 2b shows that, at various holding times, the grain growth rate of 0.65CLT-0.35SMT ceramics sintered at 1575 and 1600 °C is much faster than that at other temperatures. By comparison, sintering temperature has a more significant effect on grain growth than holding time.

SEM images of 0.65CLT-0.35SMT ceramics sintered at 1525~1600 °C for 60 min are presented in Figure 3. Strip-shaped grains can be observed in all four specimens, which is similar to the CaTiO_3_-La(Mg_0.5_Ti_0.5_) ceramics [17]. With the increasing sintering temperature, the grain growth accelerated by the increased atomic diffusion ability, which is consistent with the relationship shown in Figure 2a.

## 4. Grain Growth

### 4.1. Hillert Model

The Hillert model [18] is as follows: (1)$$d^{2}-d_{0}^{2}=At\exp[-Q/(RT)]$$

In Equation (1), *d* is the average grain size of the grains, μm; *d*_0_ is the average grain size of the original particles, 1.31 μm; *t* is the holding time, s; *T* is the sintering temperature, K; *Q* is the growth activation energy, J/mol; *R* is the gas constant; and *A* is the constant.

Equation (2) was obtained through logarithmic calculations performed on both sides of Equation (1):(2)$$\ln(d^{2}-d_{0}^{2})=\ln A+\ln t-Q/(RT)$$

To obtain *Q* and *A*, the linear relationship curves between $\ln(d^{2}-d_{0}^{2})$ and 1/T of the 0.65CLT-0.35SMT ceramics were drawn in Figure 4, according to the data in Table 1. It can be calculated that *Q* is 1,008,570 J/mol and *A* is 2.828 × 10^27^, respectively. Therefore, the Hillert grain growth model of 0.65CLT-0.35SMT ceramic is identified as Equation (3):(3)$$d^{2}-d_{0}^{2}=2.828\times 10^{27}\times t\times \exp[-1,008,570/(RT)]$$

### 4.2. Simplified Sellars Model

The Sellars model [19] is as follows:(4)$$d^{n}-d_{0}^{n}=At\exp[-Q/(RT)]$$
where *n* is the growth index. Some researchers [20,21] found that *d* is much larger than *d*_0_ (i.e., *d* >> *d*_0_) during the sintering process. Hence, the Sellars model can be simplified as Equation (5): (5)$$d^{n}=At\exp[-Q/(RT)]$$

When *T* takes a certain value, *A* is a constant. After being performed with logarithmic calculations on both sides, Equation (5) can be rewritten as:(6)nlnd=lnt+nlnA−Q/(RT)

Equation (6) can be transformed into Equation (7): (7)lnd=1/nlnt+1/nlnA−1/n[Q/(RT)]

To get *n* and *Q* of 0.65CLT-0.35SMT ceramic, the linear relationship curves of lnd~lnt and nlnd~1/T were drawn in Figure 5. Through calculation, the average growth kinetic index *n* is 3732 and the average growth activation energy *Q* is 1,865,042 kJ/mol. According to the data in Table 1, it can be calculated that *A* is 1.123 × 10^54^. Finally, the simplified Sellars model of 0.65CLT-0.35SMT ceramic grain growth leads to Equation (8): (8)$$d^{3.732}=1.123\times 10^{54}\times t\times \exp[-1,865,042/(RT)]$$

### 4.3. Sellars-Anelli Model

To establish the mathematical model of the 0.65CLT-0.35SMT ceramic grain growth during the sintering process, it is necessary to consider the influence of sintering temperature, holding time, and initial particle size on the entire system at the same time, and the growth of grain may have a certain power relationship with the holding time. The Sellars-Anelli model, Equation (9) [22], can be used to describe the process of 0.65CLT-0.35SMT ceramic grain growth.
(9)$$d^{n}-d_{0}^{n}=At^{m}\exp[-Q/(RT)]$$

In Equation (9), *d* is the average grain size of the grains, μm; *d*_0_ is the average grain size of the initial particles, μm; *t* is the holding time, s; *m* is the time index, *T* is the sintering temperature, K; *Q* is the growth activation energy, J/mol; *R* is the gas constant; *A* is the constant; and *n* is the growth index. 

In order to determine the unknown quantity of the Sellars-Anelli model, both sides of Equation (9) were performed by natural logarithm: (10)$$\ln(d^{n}-d_{0}^{n})=\ln A+m\ln t-Q/(RT)$$

To solve Equation (10), *n*, *m*, *Q,* and *A* should be determined successively. Firstly, the *n* value could be set to 0.5, 1.0, 1.5, 2.0, 2.5, 3.0, 3.5, 4.0, 4.5, 5.0, 5.5, 6.0 according to the *n* value of 3.732, which was obtained from the simplified Sellars model (Equation (7)). Secondly, the fitting relationship between $\ln(d^{n}-d_{0}^{n})$ and lnt according to Equation (10) was established to solve the *m* value and error value corresponding to each *n* value. Thirdly, the function of the sum of squared errors and each *n* value was established and then the minimum sum of squared errors was taken as the optimization goal to finally determine the value of *n*. Fourthly, the determined *n* value was returned into Equation (10). Finally, the values of *m*, *Q,* and *A* were solved in turn by establishing fitting curves of $\ln(d^{n}-d_{0}^{n})\sim \ln t$ and $\ln(d^{n}-d_{0}^{n})\sim 1/T$.

The fitting curve of the sum of squared errors with *n* is shown in Figure 6. A smooth cubic curve, $f(n)=0.010\;6n^{3}+0.268\;7n^{2}-1.875\;1n+3.306\;3$, can be obtained by fitting a cubic polynomial. It can be solved that *n* is 2.968 when the sum of squared errors is minimum.

*n* = 2.968 was substituted into Equation (10) to establish linear relationship curves of $\ln(d^{2.968}-d_{0}^{2.968})\sim \ln t$ and $\ln(d^{2.968}-d_{0}^{2.968})\sim 1/T$ for the 0.65CLT-0.35SMT ceramic, as shown in Figure 7. In Figure 7a, each curve shows the change of the grain size with the holding time at a certain sintering temperature and the *m* can be calculated as 0.889 from the average value of slope. In Figure 7b, each curve represents the change of the grain size with the sintering temperature under a certain holding time and *Q* is calculated to be 1,484,900 kJ/mol.

After *n* = 2.968, *m* = 0.889, *Q* = 1,484,900 kJ/mol was being substituted into Equation (9), *A* can be calculated as 3.936 × 10^42^ according to the measured data in Table 1. Finally, the Sellars-Anelli model for the grain growth of 0.65CLT-0.35SMT ceramic is obtained:(11)$$d^{2.968}=d_{0}^{2.968}+3.936\times 10^{42}\times t^{0.889}\times \exp[-1,484,900/(RT)]$$

### 4.4. Comparison of Different Grain Growth Models

The results predicted by the Hillard model (Equation (3)), simplified Sellars model (Equation (8)), and Sellars-Anelli model (Equation (11)) were compared, respectively, with the measured data in Table 1 to get the errors (Δ = *d*_predicted_ − *d*_measured_). The curves of $\overset{¯¯¯¯¯}{\left|\Delta \right|}$~*T* and $\overset{¯¯¯¯¯}{\left|\Delta \right|}$~*t* are shown in Figure 8a,b, respectively (where $\overset{¯¯¯¯¯}{\left|\Delta \right|}$ is the average value of the absolute value of error). The grain size predicted by the Hillert model (Equation (3)) is quite different from the measured value in Table 1, indicating the lowest prediction accuracy. By comparison, the Sellars-Anelli model (Equation (11)) is more accurate than the simplified Sellars model (Equation (8)) in the interval range because the average grain size calculated by the former is closer to the measured value. It suggests that the Sellars-Anelli model (Equation (11)) is more suitable for describing the grain growth of 0.65CLT-0.35SMT.

## 5. Conclusions


(1)The 0.65Ca_0.61_La_0.26_TiO_3_-0.35Sm(Mg_0.5_Ti_0.5_)O_3_[0.65CLT-0.35SMT] ceramic was prepared by the solid-state reaction method. The microstructure and grain growth behavior under different sintering processes were studied. The results showed that with the increase of sintering temperature and the holding time, the grain size was getting larger. Meanwhile, the effect of sintering temperature on grain growth is more obvious than that of holding time.(2)The Hillert model and simplified Sellars model of the grain growth of 0.65CLT-0.35SMT ceramic were established by the linear regression method. In addition, considering that the growth of grains may become a certain power relationship with the holding time, the nonlinear regression method was used to construct the Sellars-Anelli model with a time index.(3)An error analysis was made on the grain size predicted by the three models. The Sellars-Anelli model,$d^{2.968}=d_{0}^{2.968}+3.936\times 10^{42}\times t^{0.889}\times \exp[-1,484,900/(RT)]$, has the highest prediction accuracy, i.e., it is able to predict the grain growth process of 0.65CLT-0.35SMT ceramic effectively.


## Figures and Tables

**Figure 1 materials-13-03905-f001:**
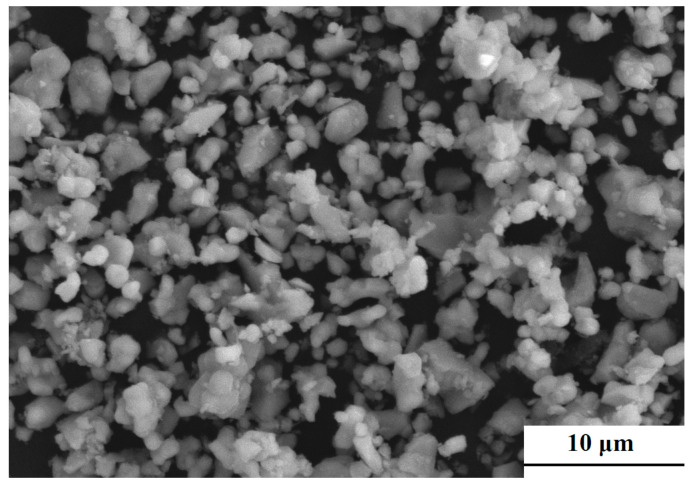
SEM image of the mixed 0.65CLT-0.35SMT powders.

**Figure 2 materials-13-03905-f002:**
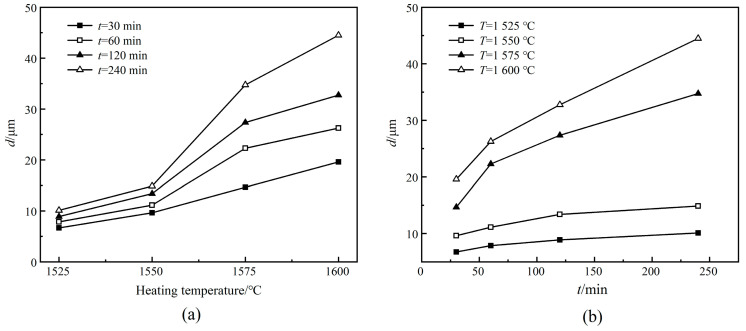
Effects of sintering temperatures (**a**) and holding times (**b**) on the average grain size of 0.65CLT-0.35SMT ceramics.

**Figure 3 materials-13-03905-f003:**
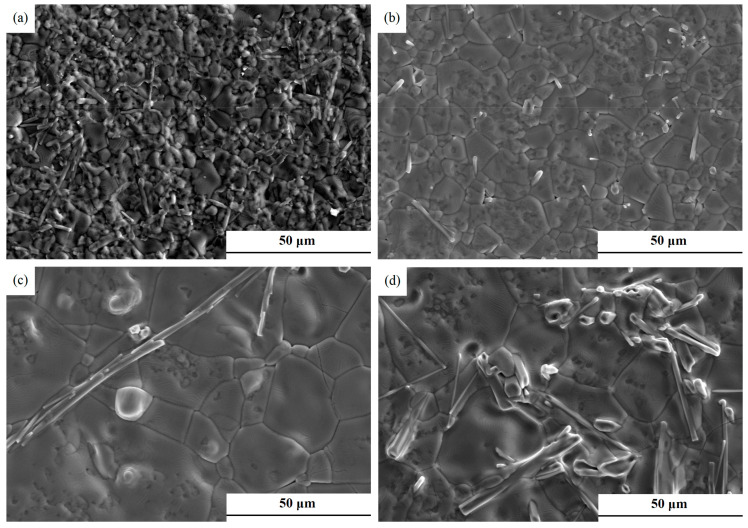
SEM images of 0.65CLT-0.35SMT ceramics sintered at different temperatures for 60 min: (**a**) 1525 °C; (**b**) 1550 °C; (**c**) 1575 °C; (**d**) 1600 °C.

**Figure 4 materials-13-03905-f004:**
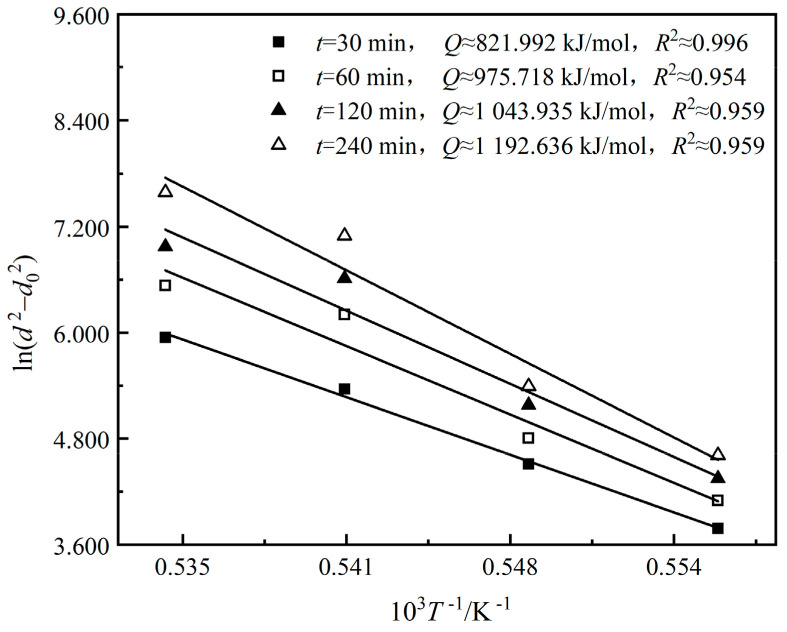
Linear relationship curve between $\ln(d^{2}-d_{0}^{2})$ and 1/T of the 0.65CLT-0.35SMT ceramics.

**Figure 5 materials-13-03905-f005:**
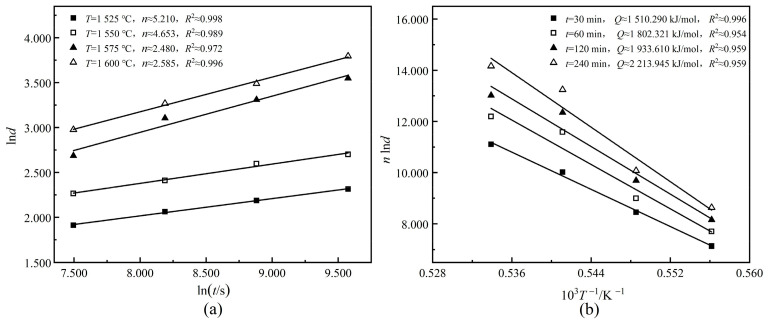
Linear relationship curves of lnd~lnt (**a**) and nlnd~1/T (**b**) for the 0.65CLT-0.35SMT ceramics.

**Figure 6 materials-13-03905-f006:**
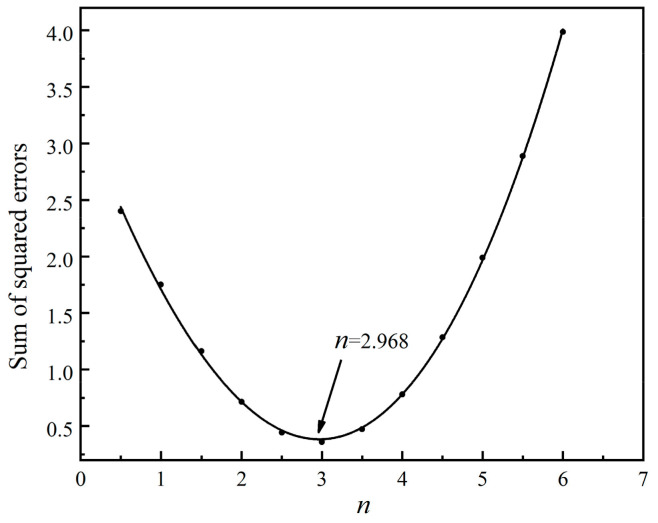
Fitted curve between the sum of squared error and *n.*

**Figure 7 materials-13-03905-f007:**
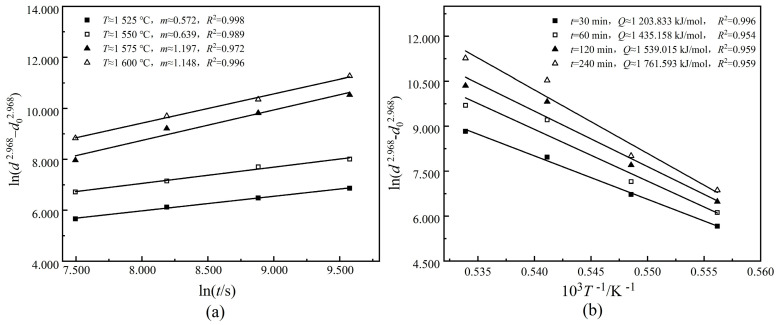
Linear relationship curves of $\ln(d^{2.968}-d_{0}^{2.968})\sim \ln t$ (**a**) and $\ln(d^{2.968}-d_{0}^{2.968})\sim 1/T$ (**b**) for the 0.65CLT-0.35SMT ceramics.

**Figure 8 materials-13-03905-f008:**
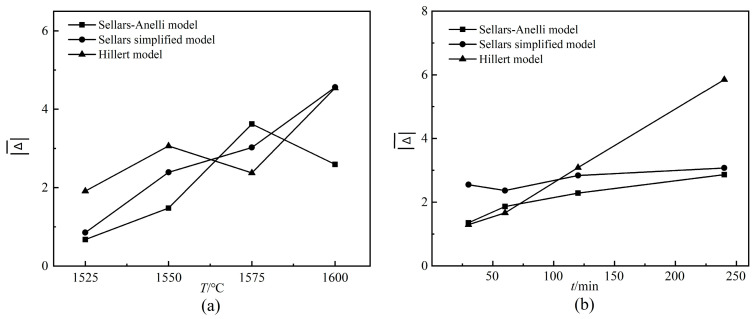
Relationship curves between $\overset{¯¯¯¯¯}{\left|\Delta \right|}$ and *T*, *t*: (**a**) Sintering temperature *T*; (**b**) holding time *t.*

**Table 1 materials-13-03905-t001:** Average grain size (*d*) and density (*η*) of 0.65CLT-0.35SMT ceramics sintered at different sintering temperatures (*T*) for different holding times (*t*).

		*t*/min	30	60	120	240
	*d*/μm(*η*/g·cm^−3^)					
*T*/°C						
1525			6.67 (5.07)	7.87 (5.08)	8.90 (4.96)	10.11 (5.00)
1550			9.63 (5.02)	11.13 (4.99)	13.41 (5.00)	14.87 (5.10)
1575			14.66 (4.98)	22.28 (5.06)	27.36 (5.06)	34.75 (5.13)
1600			19.59 (5.09)	26.25 (5.09)	32.74 (5.08)	44.49 (5.15)

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
