# Peer review of "Grain Growth Kinetics of 0.65Ca_0.61_La_0.26_TiO_3_-0.35Sm(Mg_0.5_Ti_0.5_)O_3_ Dielectric Ceramic"

_materials, 2020, doi:10.3390/ma13173905_

Round 1

Reviewer 1 Report

This work compares three different grain growth models in order to describe the grain growth in 0.65CLT-0.35SMT ceramics sintered at different temperatures and holding times. 

The work is well done and presented. It can be accepted for publication. I only have a coulpe of questions to be addressed:

  • How did the authors measure the initial powder particle size? The give a value of 1.31 μm for the average size, however, they did not describe what technique they used to obtain this value. It should be described in the Experimental Section.
  • What was the density of the sintered ceramics? Values should be included in Table 1.

Finally, I strongly recommend English to be revised.

Author Response

Ref. No.: Materials-871921Title: Grain growth kinetics of 0.65Ca0.61La0.26TiO3-0.35Sm(Mg0.5Ti0.5)O3 dielectric ceramic

Thanks to the reviewers for your time and thoughtful comments, which have been incorporated into the revised manuscript. Hopefully we have addressed all of your concerns.

Our responses to the Reviewer’s comments are presented in BOLD type as follows. The page and line numbers refer to our revised manuscript submitted at 8/24/2020.

 Reviewer #1: This work compares three different grain growth models in order to describe the grain growth in 0.65CLT-0.35SMT ceramics sintered at different temperatures and holding times. The work is well done and presented. It can be accepted for publication. I only have a couple of questions to be addressed:

  1. 1.How did the authors measure the initial powder particle size? The give a value of 1.31 μm for the average size, however, they did not describe what technique they used to obtain this value. It should be described in the Experimental Section.
  2. 1.Response: We described the measuring method of the initial powder size in the Experimental Section (in RED font, Line 70~71, Page 2). The following figure presents the measuring process.
  3. 2. What was the density of the sintered ceramics? Values should be included in Table 1. 
  4. 2.Response: The density of the sintered ceramics was listed in Table 1 (in RED font, Line 84, Page 3). 
  5. 3. Finally, I strongly recommend English to be revised.
  6. 3.Response: This manuscript has been revised carefully and thoroughly by Zhiping Li, whom is a teacher of Institute of Foreign Languages in Jiangxi Science and Technology Normal University and a Ph.D. candidate for English Language & Literature in University of Wales Trinity Saint David (2019-2022). We corrected some grammatical errors in the revised manuscript (in BLUE font). 

Reviewer 2 Report

line 46    "model"     should be replaced by     "models"

line 66    please explain what furnace, what atmosphere have been used

line 68   "After sintered samples ....."     should be replaced by text      "After it the sintered samples ...."

line 80  "ceramics at"     should be replaced by text      " "ceramics obtained at"

line 83  I propose to change the text     "an increase 97.06% from"   into the text "about 34 times greater than"

line 89   "ceramics in Figure 2."    should be replaced by text  "ceramics are presented in Figure 2." 

line 100   "in Figure 3."    should be replaced by text  " are presented in Figure 3."

line 121   what value of d0 has been taken into calculations?

line 130

line 46    "model"     should be replaced by     "models"

line 66    please explain what furnace, what atmosphere have been used

line 68   "After sintered samples ....."     should be replaced by text      "After it the sintered samples ...."

line 80  "ceramics at"     should be replaced by text      " "ceramics obtained at"

line 83  I propose to change the text     "an increase 97.06% from"   into the text "about 34 times greater than"

line 89   "ceramics in Figure 2."    should be replaced by text  "ceramics are presented in Figure 2." 

line 100   "in Figure 3."    should be replaced by text  " are presented in Figure 3."

line 121   what value of d0 has been taken into calculations?

line 130  "Perform"  should be replaced by text  "Performing"

line 131 "to obtain" should be replaced by text  "we obtain"

line 139 "is equation" should be replaced by text  "leads to equation"

line 192 please explain how the error () was calculated?

Figure 3c  and 3d   grains are not visible

Figure 4 and Figure 5   why R2 has different values? It is not the universal gas constant?

  "Perform"  should be replaced by text  "Performing"

line 131 "to obtain" should be replaced by text  "we obtain"

line 139 "is equation" should be replaced by text  "leads to equation"

line 192 please explain how the error () was calculated?

Figure 3c  and 3d   grains are not visible

Figure 4 and Figure 5   why R2 has different values? It is not the universal gas constant?

Author Response

Ref. No.: Materials-871921Title: Grain growth kinetics of 0.65Ca0.61La0.26TiO3-0.35Sm(Mg0.5Ti0.5)O3 dielectric ceramic

Thanks to the reviewers for your time and thoughtful comments, many of which have been incorporated into the revised manuscript. Hopefully we have addressed all of your concerns.

Our responses to the Reviewer’s comments are presented in BOLD type as follows. The page and line numbers refer to our revised manuscript submitted at 8/24/2020.

 Reviewer #2:  

  1. line 46: "model" should be replaced by "models"

 Response: Done (Line 45, Page 1, in RED font).

  1. line 66: please explain what furnace, what atmosphere have been used

Response: Done (Line 64~65, Page 2, in RED font).

  1. line 68: "After sintered samples ....." should be replaced by text "After it the sintered samples ...."

Response: This sentence was rewrote as “After being polished and ultrasonically cleaned, the sintered 0.65CLT-0.35SMT ceramic specimens… …” (Line 66~67, Page 2, in RED font).

  1. line 80: "ceramics at" should be replaced by text "ceramics obtained at"

Response: Done (Line 79, Page 3, in RED font).

  1. line 83: I propose to change the text "an increase 97.06% from" into the text "about 34 times greater than"

Response: Done (Line 81~82, Page 3, in RED font).

  1. line 89: "ceramics in Figure 2." should be replaced by text "ceramics are presented in Figure 2."

Response: This sentence was rewrote as “Figure 2 demonstrates the effects of sintering temperature and holding time on the average grain size of 0.65CLT-0.35SMT ceramics” (Line 86~87, Page 3, in RED font).

  1. line 100: "in Figure 3." should be replaced by text "are presented in Figure 3."

Response: Done (Line 97, Page 3, in RED font).

  1. line 121: what value of d0 has been taken into calculations?

Response: The value of d0 is 1.31 mm (Line 109, Page 4, in RED font).

  1. line 130: "Perform" should be replaced by text "Performing"

Response: Done (Line 124, Page 5, in RED font).

  1. line 131: "to obtain" should be replaced by text "we obtain"

Response: Done (Line 125, Page 5, in RED font).

11.line 139: "is equation" should be replaced by text "leads to equation"

Response: Done (Line 133, Page 5, in RED font).

  1. line 192: please explain how the error () was calculated?

Response: Done (Line 184, Page 7, in RED font).

  1. Figure 3c and 3d grains are not visible

Response: The brightness and contrast of Figure 3(c) and Figure 3(d) were adjusted (Line 101, Page 5, in RED font).

  1. Figure 4 and Figure 5: why R2 has different values? It is not the universal gas constant?

Response: Here, R2 (0<R2<1) is the goodness of fit index, instead of universal gas constant. High R2 means fit accuracy of fitting.

Reviewer 3 Report

The manuscript entitled 'Grain growth kinetics of 0.65Ca0.61La0.26TiO3-
0.35Sm(Mg0.5Ti0.5)O3 dielectric ceramics' by Liu et al. deals with applying and comparing different sintering models to the CLT-SMT ceramics. The authors tested different sintering temperatures and times, followed grain size changes and from that calculated the best fitting sintering model that could give information about the grain growth kinetics. Calculations in the scope of three chosen models seem correct and it is quite easy for the reader to follow their procedure of calculating parameters from the experimental data.

However, there are some concerns regarding the experimental procedure that lead the authors to obtain imput parameters for the proposed models. For instance, how did the authors determine the range of sintering temperatures (these are very high and special furnaces are needed to go to 1600°C). Second, there is no information about the obtained density, which is more important when monitoring the quality of sintering than grain growth. Third, heating rate plays an important role for grain growth (since lower temperatures promote grain growth, as compared to densification process that goes on at higher temperatures); grain growth is usually minimized with higher heating rates (fast firing). However, there is no information about that.

Then, since the proces is reaction sintering, did the authors check/monitor how well the two compounds are reacted? do they have any xrd data to confirm a single phase solid solution system? From SEM figures it seems that there might a liquid phase between the grains as some needle shaped crystals grew out from triple-junction points during thermal etching. The presence of a liquid phase would strongly impact grain growth so authors should keep in mind that the higher the temperature, the more likely is that there is an effect of more rapid grain growth due to the presence of a liquid. Further, the authors report on the method used for calculating grain size. However, if this is done by processing the SEM images where there are needle-like secondary phase grains growing on top, how does the software accomodate for or eliminate these grains? Also, the interior of the grains shows some decomposition - it is not clear if these are trapped pores, or an effect of thermal etching and decomposition? how does the software for grain size measurement take that into account? Finally, the authors could mention whether for given applications (that the authors describe in introduction) it is desirable to have larger or smaller grain size, since the optimum sintering is usually a compromise between optimum grains, density, phase composition etc.

All these issues may influence the sintering, grain size and the final proposed model. I suggest the authors first comment/answer these questions and/or revisit their imput data prior to publication. There are also some grammar issues (like sentences missing a verb), but these are minor issues. the term 'solid-phase reaction sintering' should be changed to 'solid-state..' There is also very little (or no) citation of the CLT-SMT system, I suggest the authors add it for the readers to be able to know more about the material.

Author Response

Ref. No.: Materials-871921Title: Grain growth kinetics of 0.65Ca0.61La0.26TiO3-0.35Sm(Mg0.5Ti0.5)O3 dielectric ceramic

Thanks to the reviewers for your time and thoughtful comments, which have been incorporated into the revised manuscript. Hopefully we have addressed all of your concerns.

Our responses to the Reviewer’s comments are presented in BOLD type as follows. The page and line numbers refer to our revised manuscript submitted at 8/24/2020.

 Reviewer #3:  The manuscript entitled 'Grain growth kinetics of 0.65Ca0.61La0.26TiO3- 0.35Sm(Mg0.5Ti0.5)O3 dielectric ceramics' by Liu et al. deals with applying and comparing different sintering models to the CLT-SMT ceramics. The authors tested different sintering temperatures and times, followed grain size changes and from that calculated the best fitting sintering model that could give information about the grain growth kinetics. Calculations in the scope of three chosen models seem correct and it is quite easy for the reader to follow their procedure of calculating parameters from the experimental data. 1. However, there are some concerns regarding the experimental procedure that lead the authors to obtain imput parameters for the proposed models. For instance, how did the authors determine the range of sintering temperatures (these are very high and special furnaces are needed to go to 1600°C). Second, there is no information about the obtained density, which is more important when monitoring the quality of sintering than grain growth. Third, heating rate plays an important role for grain growth (since lower temperatures promote grain growth, as compared to densification process that goes on at higher temperatures); grain growth is usually minimized with higher heating rates (fast firing). However, there is no information about that. Response: (1) We determine the range of sintering temperatures according to our previous work.(2) The density of the sintered ceramics was listed in Table 1 (Line 84, Page 3, in RED font).(3) The heating rate used in this work is 10 °C/min (Line 65, Page 2, in RED font).2. Then, since the process is reaction sintering, did the authors check/monitor how well the two compounds are reacted? Do they have any XRD data to confirm a single phase solid solution system? From SEM figures it seems that there might a liquid phase between the grains as some needle shaped crystals grew out from triple-junction points during thermal etching. The presence of a liquid phase would strongly impact grain growth so authors should keep in mind that the higher the temperature, the more likely is that there is an effect of more rapid grain growth due to the presence of a liquid. Further, the authors report on the method used for calculating grain size. However, if this is done by processing the SEM images where there are needle-like secondary phase grains growing on top, how does the software accommodate for or eliminate these grains? Also, the interior of the grains shows some decomposition - it is not clear if these are trapped pores, or an effect of thermal etching and decomposition? how does the software for grain size measurement take that into account? Finally, the authors could mention whether for given applications (that the authors describe in introduction) it is desirable to have larger or smaller grain size, since the optimum sintering is usually a compromise between optimum grains, density, phase composition etc. Response: (1) The XRD analysis of 0.65CLT-0.35SMT ceramic indicates that only single perovskite phase was obtained: Figure 1. XRD analysis of 0.65CLT-0.35SMT ceramic(2) Last week, we carried out the fracture morphology of 0.65CLT-0.35SMT (1600 °C, 240min) ceramic by SEM. The result shows that no needle shaped crystal was found in the internal part of this ceramic.(3) Response: We described the measuring method of the initial powder size in the Experimental Section (in RED font, Line 70~71, Page 2). All the size data were measured manually one by one. The following figure presents the measuring process. 3. All these issues may influence the sintering, grain size and the final proposed model. I suggest the authors first comment/answer these questions and/or revisit their imput data prior to publication. There are also some grammar issues (like sentences missing a verb), but these are minor issues. The term 'solid-phase reaction sintering' should be changed to 'solid-state...' There is also very little (or no) citation of the CLT-SMT system, I suggest the authors add it for the readers to be able to know more about the material. Response: (1) This manuscript has been revised carefully and thoroughly by Zhiping Li, whom is a teacher of Institute of Foreign Languages in Jiangxi Science and Technology Normal University and a Ph.D. candidate for English Language & Literature in University of Wales Trinity Saint David (2019-2022). We corrected some grammatical errors in the revised manuscript (in BLUE font).(2) The term “solid-phase” has been changed to “solid-state” (Line 13, Page 1; Line 196, Page 8, in RED font).(3) As far as we know, no paper about CLT-SMT published yet. So we added the citation about the similar system, such as CLT-LMT and CST-LMT. (Ref.[11] and Ref.[12]; Line38, Page 1; Line 240-244, Page 9; in RED font). Of course, the number of the citations was updated also.
